# Maternal Mental Health Symptom Profiles and Infant Sleep: A Cross-Sectional Survey

**DOI:** 10.3390/diagnostics12071625

**Published:** 2022-07-04

**Authors:** Vania Sandoz, Alain Lacroix, Suzannah Stuijfzand, Myriam Bickle Graz, Antje Horsch

**Affiliations:** 1Institute of Higher Education and Research in Healthcare, University of Lausanne, 1010 Lausanne, Switzerland; vania.sandoz@bluewin.ch (V.S.); alain.lacroix@bluewin.ch (A.L.); suzannah.stuijfzand@protonmail.com (S.S.); 2Child Abuse and Neglect Team, Department Woman-Mother-Child, Lausanne University Hospital, 1011 Lausanne, Switzerland; 3Neonatology Unit, Department Woman-Mother-Child, Lausanne University Hospital, 1011 Lausanne, Switzerland; myriam.bickle-graz@chuv.ch

**Keywords:** infant sleep, depression, PTSD, anxiety, birth trauma, mothers, temperament, HADS, EPDS, City BiTS

## Abstract

The distinct influence of different, but comorbid, maternal mental health (MMH) difficulties (postpartum depression, anxiety, childbirth-related posttraumatic stress disorder) on infant sleep is unknown, although associations between MMH and infant sleep were reported. This cross-sectional survey aimed: (1) to examine associations between MMH symptoms and infant sleep; (2) to extract data-driven maternal MMH symptom profiles from MMH symptoms; and (3) to investigate the distinct influence of these MMH symptom profiles on infant sleep when including mediators and moderators. Mothers of 3–12-month-old infants (*n* = 410) completed standardized questionnaires on infant sleep, maternal perception of infant negative emotionality, and MMH symptoms. Data was analyzed using: (1) simple linear regressions; (2) factor analysis; and (3) structural equation modelling. MMH symptoms were all negatively associated with nocturnal sleep duration and only postpartum depression and anxiety symptoms were associated with night waking. Three MMH symptom profiles were extracted: depressive, anxious, and birth trauma profiles. Maternal perception of infant negative emotionality mediated the associations between the depressive or anxious profiles and infant sleep but only for particular infant ages or maternal education levels. The birth trauma profile was not associated with infant sleep. The relationships between MMH and infant sleep may involve distinct mechanisms contingent on maternal symptomatology.

## 1. Introduction

Although sleep problems (e.g., short nocturnal sleep duration or night waking) during infancy are expected to vary among infants and over time as part of their development [1], the prevalence of sleep problems during infancy is estimated to be between 10% and 17% [2,3]. Mothers typically reported short nocturnal sleep duration and/or night waking [2,3]. Focusing on sleep problems during infancy and childhood is important because they are negatively associated with maternal well-being [4,5] and infant developmental outcomes [6]. Interestingly, 69% of child insomnia does not have a physical cause, suggesting that it may be linked to parental behavior [7], e.g., nighttime parenting has been shown to be associated with infant night waking [4]. Thus, several maternal nighttime behaviors may interfere with the infant’s acquisition of self-regulation skills necessary to maintain sleep through the night or to self-soothe back to sleep [8]. Associations between maternal nighttime parenting and maternal postpartum depression symptoms have also been reported [8].

Postpartum depression is common following childbirth, as are anxiety and childbirth-related posttraumatic stress disorder (CB-PTSD) [9,10,11,12,13]. Postpartum depression is characterized by a general state of low mood and/or anhedonia and affects 13% of mothers during the first year postpartum [9,10,14]. Anxiety involves worries, avoidance, obsessions, and physical changes, and is prevalent in up to 15% of mothers during the first six months postpartum [10,11,14]. Finally, CB-PTSD develops after traumatic childbirth (i.e., perceived threat to the life of the mother and/or infant) and contains symptoms of re-experiencing, avoidance, negative cognitions and mood, and hyperarousal [14,15]. The prevalence of maternal CB-PTSD is estimated at 3–4% in low-risk samples and 16–19% in high-risk populations [12,13].

Mounting evidence shows associations between maternal symptoms of postpartum depression or anxiety and the sleep of their offspring [4,5,16,17,18]. However, regarding causal associations, findings of the few prospective studies are inconsistent, although recent results support mother-driven mechanisms in infancy [5,16,17]. Moreover, the influence of maternal CB-PTSD symptoms on infant sleep during the first year postpartum has never been investigated, although maternal CB-PTSD symptoms at 8 weeks postpartum were shown to be prospectively linked to child sleep at 2 years postpartum [19].

The transactional model of infant sleep and parenting describes the interactions involved in infant sleep [8,20]. This model proposes that infant sleep shares complex and ongoing relationships with (1) the distal extrinsic context (e.g., maternal education), (2) the parenting factors context (e.g., MMH), (3) intrinsic infant factors (e.g., infant age), and (4) the parent–infant interactive context, including interpersonal systems (e.g., maternal perception of infant temperament) and interactive behaviors factors (e.g., bedtime interactions) [8,20]. The evidence so far has reported associations between infant sleep, perinatal MMH symptoms (mainly depression), infant temperament (e.g., negative emotionality, often referred to as negative affectivity or negative temperament), and bedtime interactions (e.g., method of falling asleep), therefore supporting this theoretical framework [8,21,22,23,24]. Note that these studies diverge by their participants’ age, variables of interest, study designs (e.g., cross sectional or longitudinal), as well as assessment tools and time points for their offspring sleep and MMH. Importantly, findings on mechanisms underlying these associations are inconsistent and additional research is therefore required, e.g., on the role of maternal perception of infant temperament in the relationships between MMH and infant sleep.

### 1.1. Comorbidity of Maternal Mental Health Difficulties and Their Influence on Infant Sleep

Given the important comorbidity between maternal postpartum depression, anxiety, and CB-PTSD [14,25,26,27], it raises the question of whether these mental health difficulties are distinct childbirth-related phenotypes. For example, in a previous cross-sectional study, postpartum depression and CB-PTSD symptoms loaded onto one-factor model explaining 68% of the total variance, when excluding the overlapping symptoms (i.e., anhedonia, sleep disturbance, and concentration difficulties) [25]. Moreover, risk factors involved in comorbid postpartum depression and CB-PTSD were different from the ones associated with postpartum depression or CB-PTSD alone [25]. Authors, therefore, suggested a specific posttraumatic stress-depressive profile that differs from postpartum depression or CB-PTSD [25].

The comorbidity between MMH difficulties during the postpartum period also raises the question of how they interact together to influence infant sleep. To better understand the relationships between these postpartum disorders and infant sleep is important for both research and clinical practice. From a research point of view, if postpartum depression, anxiety, and CB-PTSD are associated with infant sleep via shared mechanisms, they could be investigated as one concept, whereas if distinct processes are involved, they would have to be considered separately. From a clinical point of view, this could have major implications on how families are cared for by perinatal health professionals, as mothers seldom experience symptoms of postpartum depression, anxiety, or CB-PTSD alone. To our knowledge, the shared influences of distinct mechanisms of postpartum depression, anxiety, and CB-PTSD contributing to the deleterious effects on infant sleep have never been examined so far.

### 1.2. The Current Study

The first purpose of the current study was to examine the associations between maternal symptoms of postpartum depression, anxiety, or CB-PTSD and infant sleep (i.e., night waking and nocturnal sleep duration). Based on the literature, we hypothesized that MMH symptoms are positively associated with infant night waking, but negatively with nocturnal sleep duration [4,19,28]. Given that postpartum depression, anxiety, and CB-PTSD are comorbid [14,25,26,27], the second aim consisted of extracting data-driven MMH symptom profiles. We had no specific assumption, as this was exploratory. Finally, the third aim was to investigate the distinct influence of these MMH symptom profiles on infant sleep when including mediators (i.e., the method of falling asleep and maternal perception of infant negative emotionality) and moderators (i.e., maternal education or infant age). Our hypothesis regarding the pathways to be tested were based on the transactional model of infant sleep and parenting and can been seen in Figure 1 [8,20].

## 2. Materials and Methods

### 2.1. Study Design and Population

This online cross-sectional study included 410 mother–infant dyads. Mothers and birth partners (i.e., male or female co-parent) first participated in a validation study and then completed an optional part assessing infant sleep and temperament. Details of the maternal validation study (*n* = 541) are reported elsewhere [29]. Given that data collection for the birth partners is still ongoing, they will also be reported elsewhere. Therefore, the current paper only includes mothers who completed the two parts of the survey. Maternal eligibility criteria consisted of being the birth mother of an infant aged 3 to 12 months old, being ≥18 years old, and speaking French. For the 1.2% of participants who had twins, only data related to the first-born baby was used in the current study.

### 2.2. Infant and Maternal Measures

*Brief Infant Sleep Questionnaire:* This maternal-report questionnaire assessed various infant sleep variables over the previous week, such as nocturnal sleep duration (between 7 p.m. to 7 a.m.), number of night waking, and method of falling asleep with the following response options: while being fed = 1, while being rocked = 2, while being held = 3, alone in the crib = 4, and in the crib with parental presence = 5 [30]. A French translation and cultural adaptation was carried out according to the forward–backward method [31]. The original study demonstrated good psychometric properties for this questionnaire, including test–retest reliability and clinical validity [30].

*Negative Emotionality dimension of the Very Short Form of the Infant Behavior Questionnaire-Revised (IBQ-NEG):* This maternal-report questionnaire measured maternal perception of infant temperament, including the frequency of recent and concrete infant behaviors reported by mothers on a 7-point Likert scale [32]. One of the three dimensions assessed by this maternal-report questionnaire is negative emotionality (IBQ-NEG). The IBQ-NEG 12-items indicate the tendency for the infant to express negative emotions, such as sadness, distress to limitation, and fear [32]. The total score ranges from 1 to 7, with a higher score indicating higher negative emotionality [32]. Given that no validated French version exists, the forward–backward method was used for cultural adaptation and French translation [31]. Good psychometric properties were reported for this questionnaire [32], and the IBQ-NEG internal consistency in this study was adequate (Cronbach’s α = 0.82).

*Edinburgh Postnatal Depression Scale (EPDS):* This 10-item self-report questionnaire assessed maternal postpartum depression symptoms within the last week [33]. A higher total score (range: 0–40) indicates higher symptom severity [33]. The French version of the EPDS showed good psychometric characteristics [34]. In the current study, internal consistency was appropriate (Cronbach’s α = 0.80) and slightly lower than that previously reported in postpartum French-speaking mothers of Switzerland [35].

*Anxiety Subscale of the Hospital Anxiety and Depression Scale (HADS-A):* Anxiety symptoms occurring in the last week were assessed with the HADS-A [36]. The total score of this 7-item self-report questionnaire ranges from 0 to 21, with higher scores suggesting higher symptom severity [36]. Good psychometric properties have been reported in the French version [37]. In the current study, Cronbach’s α was good at 0.90, which was higher than what was previously reported in postpartum French-speaking mother of Switzerland [38].

*City Birth Trauma Scale (City BiTS):* The City BiTS is a self-report tool measuring the frequency of CB-PTSD symptoms over the last month [15]. The City BiTS contains 29 items, with 20 of them evaluating PTSD symptom clusters of the Diagnostic and Statistical Manual of Mental Disorders, 5th ed. (DSM-5; criteria B to E), namely intrusions, avoidance, negative cognitions and mood, and hyperarousal [14,15]. The City BiTS contains the birth-related symptoms subscale and the general symptoms subscales [15]. The birth-related symptoms subscale is composed of items assessing symptoms of intrusion, avoidance, and a few that measure negative cognitions and mood. The general symptoms subscale consists of the rest of the items assessing negative cognitions and mood, and hyperarousal symptoms. Greater severity of CB-PTSD symptoms is suggested by a higher total score, which includes DSM-5 criteria B-E items (range: 0–60). The French version has demonstrated good psychometric properties [29]. The Cronbach’s α in the current study was good at 0.82, which was slightly lower than what was observed in postpartum English-speaking mothers [29].

*Sociodemographic and medical data:* The following information was collected via single items completed by the mothers: maternal age, marital status, and educational level (no education = 1, compulsory education = 2, post-compulsory education = 3, university of applied science or university diploma of technology degree = 4, and university = 5), as well as weeks of gestation, and infant gender and age (≥3 months to <6 months = 1, ≥6 months to <9 months = 2, and ≥9 months to <12 months = 3).

### 2.3. Procedure

This online study was hosted on Sphinx iQ2, allowing data to be stored on a secure server owned by a Swiss university hospital. Data was collected between June and September 2020. The study was advertised mainly via social media, such as Facebook and Instagram, but also via personal and professional networks and nurseries. Given that participants had to complete the last page of the survey for data to be saved, no information was available for early dropouts. The local ethics committee classified the study as anonymous, therefore not requiring full approval processing. Data are available free of charge and without restriction from the open access repository Zenodo (https://doi.org/10.5281/zenodo.5070945) [39].

### 2.4. Statistical Analysis

Descriptive analyses and simple linear regressions were performed with IBM SPSS Statistics 27 (SPSS Inc., Chicago, IL, USA), while the rest of the analyses were carried out with R studio version 1.2.5033 (RStudio Team, 2020 http://www.rstudio.com/ (accessed on 4 May 2022)) and R version 3.6.2 (R Core Team, 2021 https://www.R-project.org/ (accessed on 4 May 2022)). Descriptive and exploratory analyses were conducted first to ensure that the data were appropriate for the planned analyses.

To detect relationships between maternal symptoms of postpartum depression, anxiety, or CB-PTSD symptoms and infant sleep (i.e., nocturnal sleep duration and night waking), six simple linear regressions were conducted. Maternal EPDS, HADS-A, or City BiTS score were used as predictors; infant nocturnal sleep duration or night waking were dependent variables.

To establish MMH symptom profiles, an exploratory factor analysis with three predefined factors was conducted. The factor scores were computed as the sum of the items with a factor loading greater than 0.40. The main objective that drove this approach was the simplicity of the interpretation of the computed scores. Cronbach’s α was used to assess internal consistency. A confirmatory factor analysis was then conducted to assess the quality of the model. The following fit indices were used to evaluate the fit to the data: root mean square error of approximation (RMSEA), comparative fit index (CFI), Tucker–Lewis index (TLI), and standardized root mean square residual (SRMR). As a non-significant χ2 test indicating a good fit is rarely obtained with a large data set [40], we used the statistic adjusted by its degrees of freedom instead. RMSEA values below 0.06, CFI and TLI values above 0.95, and SRMR values below 0.08 indicate a good fit [41]. It has been suggested, with some consensus in the psychometric literature, that a model demonstrates reasonable fit if the value of χ2/df does not exceed 3.0 [42]. No outliers were detected.

Finally, to take into account the role of mediators (i.e., IBQ-NEG and method of falling asleep) and moderators (i.e., maternal educational level and infant age) in the associations between MMH symptom profiles and infant sleep, we first recoded the educational level and method of falling asleep into dichotomous variables. Low educational level included responses *1*, *2*, and *3*, whereas high educational level comprised responses *4* and *5*. Regarding the method of falling asleep, responses *1, 2, 3*, and *5* were grouped as interactive method of falling asleep, while the non-interactive method of falling asleep only included response *4*. This choice was theory-driven, since infants who fall asleep with minimal parental involvement (e.g., alone in their crib) were reported to have better sleep outcomes [8]. Mediation and moderation effects have been tested using structural equation modelling. To test for the significance of the indirect effects within the mediation models, bootstrapping with 1000 iterations was used. To assess statistical power, post hoc analyses were conducted for the testing mediation effect based on Sobel’s test and Monte Carlo simulations in case the indirect effect was not normally distributed [43,44].

Due to a technical issue, the response of one participant was missing for nocturnal sleep duration, which was not imputed. Results were considered as significant at *p* < 0.05.

## 3. Results

### 3.1. Characteristics of the Sample

Mothers were on average 30.20 years old (*SD* = 4.36), and most of them (94.9%) reported being in a couple relationship. Concerning infants, 51.7% were female and their age was, to a certain extent, equally distributed between categories (Table 1). Regarding the method of falling asleep, 56.8% children fell asleep with an interactive method, while a 43.2% required a non-interactive method of falling asleep. In addition, 31.7% of mothers had a low educational level and 68.3% reported a high educational level. More information on the characteristics of the sample are displayed Table 1.

### 3.2. Associations between Maternal Mental Health Symptoms and Infant Sleep Problems

Associations between MMH symptoms (i.e., total score of EPDS, HADS-A, and City BiTS) and infant sleep (i.e., night waking and nocturnal sleep duration) are shown in Table 2. All associations tested were significant, except for the association between City BiTS total score and night waking.

### 3.3. Maternal Mental Health Symptom Profiles

Loading values > 0.40 resulting from the exploratory factor analysis with three predefined factors are displayed in Table 3. The quality indices showed an acceptable fit to the data (*RMSEA* = 0.074, *CFI* = 0.862, *TLI* = 0.851, *χ2/df* = 3.238, *SRMR* = 0.056). Eight EPDS items, two HADS-A items, and eight City BiTS items loaded on the first factor, which was named the *depressive profile*. Only nine City BiTS items, all of which were from the birth-related symptoms subscale, loaded on the second factor, entitled the *birth trauma profile*. Finally, three EPDS items, four HADS-A items, and two City BiTS items loaded on the third factor that was called the *anxious profile*. Item 11 of the HADS-A and items 8 and 18 of the City BiTS were included in neither of these three latent factors, since their loading values were <0.40. The depressive profile was highly correlated with the anxious profile (*r* = 0.81, *p* < 0.001) but moderately with birth trauma profile (*r* = 0.41, *p* < 0.001). The anxious profile was also moderately correlated with the birth trauma profile (*r* = 0.46, *p* < 0.001).

### 3.4. Associations of Maternal Mental Health Symptom Profiles to Infant Sleep Problems

Out of the twenty-four moderated mediation models, five were significant (Table 4). Non-standardized beta coefficients are displayed in Figure 2. The association between the depressive profile and night waking was mediated by IBQ-NEG, whatever the infant age (Model 1) or maternal education level (Model 2). The association between the depressive profile and sleep night duration was also mediated via IBQ-NEG for infants aged 6 to 9 months (Model 3). Finally, IBQ-NEG mediated the association between the anxious profile and night waking, but only when the infant age was between 3 to 6 months (Model 4) or maternal educational was high (Model 5). Statistical information on the non-significant moderated mediation models is reported in the Appendix A. Post hoc analyses are shown in Table 4 and Appendix A.

## 4. Discussion

To our knowledge, this is the first study to explore the distinct effects of different, but comorbid, MMH difficulties (postpartum depression, anxiety, and CB-PTSD) on infant sleep in the first year postpartum. Maternal postpartum depression and anxiety symptoms were associated with more night waking and less nocturnal sleep duration, as reported in previous studies [4,5], whereas CB-PTSD symptoms were only associated with a shorter nocturnal sleep duration. When analyzing the EPDS, HADS-A, and City BiTS items, three mental health symptom profiles emerged, namely the depressive, the birth trauma, and the anxious profiles. The mothers’ perception of infant negative temperament mediated several associations. Hence, the maternal perception of infant negative emotionality mediated the association between the depressive profile and night waking, whatever the infant age (Model 1) or maternal education (Model 2); in 6–9 months old infants the association between depressive profile and nocturnal sleep duration (Model 3); and finally, the association between the anxious profile and night waking when infants were 3 to 6 months old (Model 4) or maternal education was high (Model 5). We found no mediating effect of infant temperament perception on the association between the birth trauma profile and infant sleep. Finally, the method of falling asleep did not mediate the influence of any of the MMH symptom profiles on infant sleep problems, contrary to what was expected based on the transactional model of infant sleep and parenting [8].

In contrast to assumptions, maternal symptoms of CB-PTSD were not linked to infant night waking. One explanation may be that the infant may act as a reminder of the trauma, which could trigger the re-experiencing of symptoms [45,46]. As a result, mothers may experience infant-related avoidance (PTSD criteria C) and require their partner to care for the infant at night [45,46]. This is supported by findings demonstrating that mothers with CB-PTSD symptoms showed infant-directed hostility and diminished pleasure when interacting with their infant [45]. Moreover, mothers with a likely CB-PTSD diagnosis showed less desire for proximity to their offspring [45]. This assumption is speculative though, since partner involvement was not assessed in our study.

The unexpected lack of association between maternal CB-PTSD symptoms and night waking contrasts with the significant negative relationship observed between maternal CB-PTSD symptoms and nocturnal sleep duration. CB-PTSD-specific mechanisms associated with infant sleep may differ from the ones related to postpartum depression or anxiety. Given that CB-PTSD symptoms can impact maternal infant perception [45], mothers with CB-PTSD symptoms could have distorted beliefs concerning the expected night waking in infants, which might influence their reporting; however, this remains to be investigated. Since maternal CB-PTSD symptoms at 8 weeks postpartum prospectively predicted child sleep problems at two years postpartum [19], this suggests that the consequences of the intergenerational transmission of stress and trauma on infant sleep may develop after the first year postpartum.

Three profiles of mental health symptoms in mothers during the postpartum period emerged, namely the depressive, birth trauma, and anxious profiles. The depressive profile was characterized by symptoms linked to low mood, anhedonia, concentration problems, guilt, anger, irritability, social detachment, and self-destruction. The birth trauma profile covered symptoms including intrusive memories, avoidance, and negative mood and cognitions, all of which are related to childbirth. Finally, in the anxious profile symptoms of excessive worrying, infant-unrelated sleep difficulties, panic, and fear were present. Both the depressive and anxious profiles were composed of various symptoms of postpartum depression, anxiety, and CB-PTSD, while the birth trauma profile contained solely birth-related symptoms of CB-PTSD. In line with previous findings [25], this may suggest that these mental health difficulties are not totally distinct childbirth-related phenotypes. Given the important comorbidity of these disorders [14,25,26,27], adopting a holistic approach and using MMH symptom profiles can be of relevance for both clinical and research practice.

Three items of the City BiTS and HADS-A questionnaires did not contribute to the mental health profiles, which is relevant from a theoretical point of view. City BiTS item 8 relates to the difficulty of remembering the birth. Unlike other traumatic events, mothers usually experience childbirth with a birth partner, who can remind them about their experience. City BiTS item 18 refers to hypervigilance and HADS-A item 6 to feeling restless, and both could be seen as the norm when taking care of an infant.

Finally, an important aspect of the current study is the exploration of the distinct influence of the three profiles of MMH symptoms on infant sleep problems. Both the associations between the depressive or anxious profiles and infant sleep were mediated by maternal perception of infant negative emotionality. Maternal symptoms of postpartum depression and anxiety have been shown to be associated with maternal perception of infant negative emotionality [47]. Moreover, anxious or depressed mothers tended to report more a difficult infant temperament, or to show more hostile feelings towards their infant and a more negative perception of their behavior in comparison with controls, respectively [48,49]. This negative perception of the infant may be linked to a behavior less attuned to their infant in depressed or anxious mothers, interfering with the acquisition of self-regulation skills [50]. Thus, the associations between maternal depressive or anxiety symptoms and infant sleep may share common mechanisms, as illustrated by the high correlations observed between the depressive and anxious profiles.

It is important to note that the mediating effect in mothers with an anxious profile of the perception of negative infant temperament on night waking was only significant when maternal education was high or for infants aged 3 to 6 months old. A study showed that 4-month-old infants of mothers with a panic disorder woke up more often than controls [18]. Their mothers were less sensitive with them and displayed parenting behaviors that may interfere with infant sleep (e.g., increased feeding at night) [18]. The longest self-regulated sleep period (i.e., time of behavioral quietude with sleep and calm awakening) starts stabilizing around 4 months [51]. It is therefore likely that infants aged between 3 to 6 months begin to sleep at night without manifesting distress, since they can self-soothe alone [51]. In anxious mothers, who have a negative infant perception, hypervigilant behaviors towards their infant might develop as a response to this new sleep rhythm. For example, mothers, who are used to repeatedly waking up to help their infant fall back to sleep, may need to alleviate their anxiety by ensuring that their infant is healthy when they start sleeping through the night, which may inadvertently lead to maternal intrusive behavior [4]. This intrusive maternal behavior may, in turn, impede infant sleep [5].

The finding related to maternal education was surprising. A higher educational level may give indications on the socioeconomic environment that is associated with parental cognitions and style, which are related to infant sleep [8]. This remains to be explored in more detail in further studies.

Maternal perception of negative infant temperament mediated the association between the depressive profile and nocturnal sleep duration, but only for infants aged between 6 to 9 months. By the age of 6 months, infants are expected to sleep throughout the night [52]. Teti and Crosby described behaviors in mothers with high depression symptoms who had concerns regarding infant needs at night that could result in infant waking [4]. Authors proposed that mothers would engage in behaviors at night that would not aim to soothe their infant but rather to bring her closer to them (e.g., nursing them without sign of hunger, picking up their sleeping infant) [4]. Their primary goal would then be to satisfy their own emotional needs [4]. Furthermore, the possible cessation of breastfeeding occurring at this time (i.e., the rates of breastfeeding fall from 60% at 2 months postpartum, to 42% at 6 months postpartum, and 24% at 12 months postpartum [53]) might also explain this result, since breastfeeding is a soothing method [52]. Thus, depressive mothers with a negative infant perception may struggle to soothe their infant with less than maximal maternal support (e.g., rocking their infant) [52], which, in turn, may interfere with the infant acquiring self-regulation competencies [8]. This hypothesis remains to be tested though.

Finally, it is interesting to note that the effect of the birth trauma profile on infant sleep was not mediated by the mother’s perception of negative infant emotionality. Our findings, therefore, suggests that different MMH symptoms influence infant sleep through distinct mechanisms. Recent results observed a lack of association between maternal birth-related symptoms of CB-PTSD and bonding, while general symptoms of CB-PTSD had a direct effect on bonding and an indirect effect via depressive symptoms [54]. This suggests that the two dimensions of CB-PTSD, namely the birth-related symptoms and general symptoms, may have distinct influence on infant outcomes.

### Strengths and Limitations

This study addressed several gaps in the literature in perinatal research. To our knowledge, this is the first study to examine maternal CB-PTSD symptoms in relation to infant sleep. Secondly, this study considered the high comorbidity between the different MMH difficulties and explored for the first time the association between distinct MMH symptom profiles and infant sleep. Third, the large sample size allowed the use of advanced analyses controlling for moderators and mediators, maximizing the potential insights that could be gained from the current study. Finally, the current study has important clinical and research implications. Our findings suggest that the maternal negative perception of infant temperament must be considered in patients with infant sleep problems, as future infant sleep interventions could target this perception. From a research perspective, future studies should examine the mechanisms involved in the associations between MMH symptom profiles, infant sleep, maternal perception of infant temperament, socioeconomic factors, and intrinsic infant factors.

Nonetheless, some limitations must be highlighted. First of all, information on current breastfeeding status and partner involvement or mental health was not collected. The role of these variables in the relationship between MMH and infant sleep could thus not be examined in the present analyses. Second, given the study design (i.e., online cross-sectional study), no physiological measure of sleep was collected, and data were self-reported. This could have introduced a self-report bias since a recent study reported a discrepancy between the BISQ when completed by mothers and actigraphy [55]. Hence, one must be cautious with the interpretation of the current results, although it has been reported that the number of infant night waking reported by mothers was not biased by maternal postpartum depression [4]. Future research should therefore use objective measures (e.g., actigraphy, clinical interviews, or behavioral assessment) not only in infants but also in both parents during the pregnancy and postpartum period [8]. This leads to the third limitation of this study, which was the lack of MMH measurement during pregnancy. Indeed, antenatal maternal depression symptoms were prospectively associated with infant sleep [16,28]. Thus, the current study would have benefitted from controlling for antenatal MMH symptoms. Fourth, for statistical purposes, the method-of-falling-asleep variable was grouped into a dichotomous variable. Hence, the sensitivity of this measure may have been impacted while performing this statistical manipulation, which could potentially explain that the method of falling asleep did not mediate any relationships between MMH symptom profile and infant sleep. Fifth, although the sample size was large enough to obtain acceptable statistical power (i.e., >0.80) when the mediation effects were significant and when the mediation effects were small and not significant, the corresponding statistical power was not large enough to correctly reject the null hypothesis. Therefore, for future research, increasing the sample size should be envisaged. Finally, no information was collected on the reasons for the choice of the method of falling asleep used by mothers, nor on maternal cognitions linked to their parenting style during the night. Such information would definitively be interesting in future research to examine the influence of MMH on infant sleep.

## 5. Conclusions

The current study investigated for the first time the distinct influences of different, but comorbid, maternal mental health symptoms on infant sleep. Findings showed that birth trauma symptoms (e.g., birth-related flashbacks or avoidance) were not associated with any of the infant sleep outcomes (i.e., nocturnal sleep duration and night waking). In contrast, the influence of the depressive profile (e.g., anhedonia, low mood or culpability) or the anxious profile (e.g., abnormal or constant worries) on infant sleep were mediated by the maternal perception of infant negative emotionality, but only for some specific infant ages and maternal educational levels. These results may indicate that different mechanisms are engaged in the associations between MMH and infant sleep, depending on the type of maternal symptomatology. The consequences of the maternal childbirth-related trauma on infant sleep may need a longer follow-up for associations to unfold, contrary to the depressive or anxious context. Future research should use longitudinal designs to determine the nature of the relationships between MMH and infant sleep, with the assumption that, first, these associations are parent-driven, and, in a second step, may become infant-driven when infant sleep problems are chronic and prolonged, resulting in bidirectional associations [4,56,57]. Finally, additional studies with physiological and psychological measures in both parents during pregnancy and postpartum, and with objective sleep measures, such as an actigraphy, are needed [8].

## Figures and Tables

**Figure 1 diagnostics-12-01625-f001:**
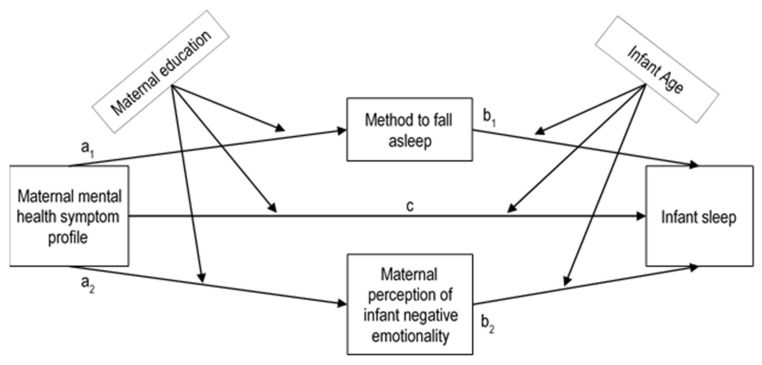
Tested pathways playing a role in the association between a maternal mental health symptom profile and an infant sleep outcome. The coefficients a_1_ and a_2_ are the coefficients of the two linear relationships between each mediator and the independent variable, while the coefficients b_1_, b_2_ and c are the coefficients of the linear relationship between the dependent variable, the mediators and the independent variable.

**Figure 2 diagnostics-12-01625-f002:**
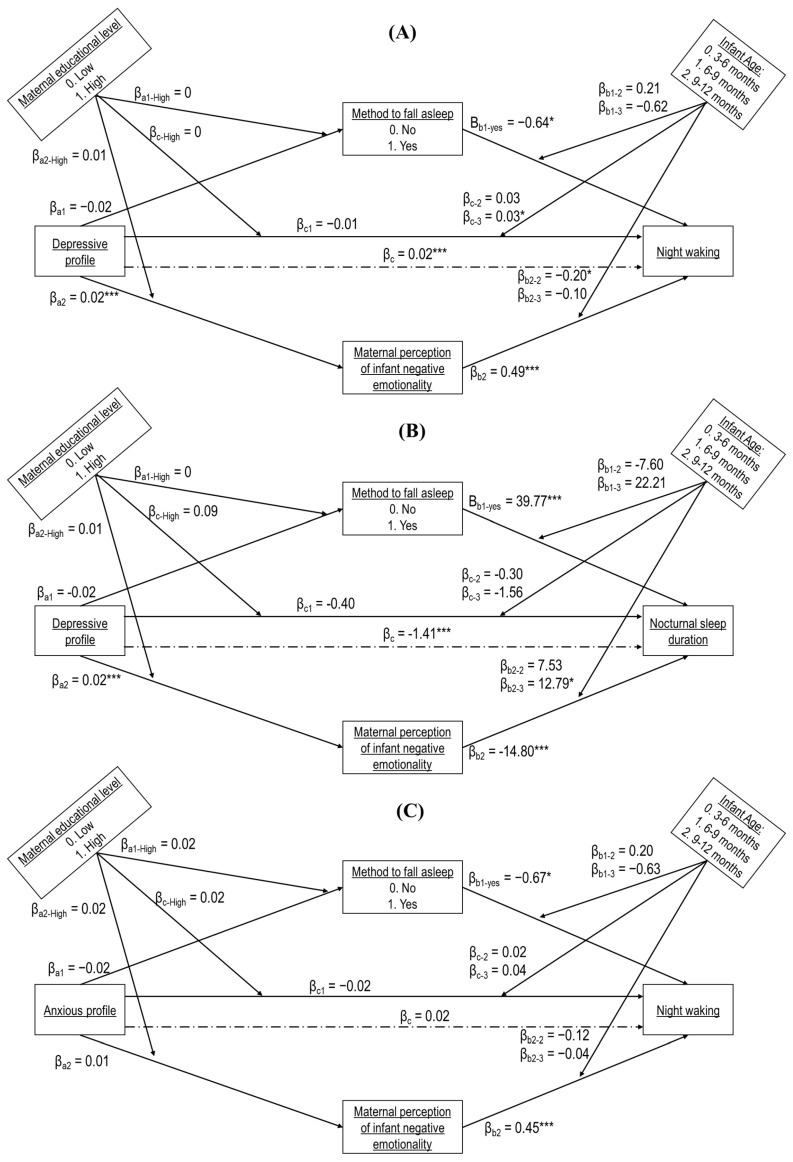
Model (**A**) displays the path model of the effect of the depressive profile on night waking, including mediators and moderators. Model (**B**) presents the path model of the effect of nocturnal sleep duration by the depressive profile, taking into account mediators and moderators. Model (**C**) represents the path model of the effect of the anxious profile on night waking, including mediators and moderators. Maternal perception of infant negative emotionality and the method of falling asleep are tested as mediators for all models, as well as maternal educational level and infant age as moderators. Dashed lines show the direct associations between the maternal mental health symptom profiles and infant sleep indicators, without including the other factors in the model. Non-standardized beta coefficients are reported. * *p* < 0.05; *** *p* < 0.001.

**Table 1 diagnostics-12-01625-t001:** Descriptive Characteristics of the Sample.

Variables	Participants (*n* = 410)	
	M (SD)	*n* (%)
**Maternal age**	30.20 (4.36)	
**Educational level**		
No education		2 (0.5)
Compulsory education		25 (6.1)
Post-compulsory education (e.g., apprenticeship)		103 (25.1)
University of Applied Science or University Diploma of Technology Degree		88 (21.5)
University		192 (46.8)
**Marital status**		
Single		14 (3.4)
In a couple relationship		389 (94.9)
Separated, divorced, or widowed		7 (1.7)
**EPDS total score**	9.05 (6.76)	
**HADS-A total score**	7.84 (4.26)	
**City BiTS total score**	13.12 (10.81)	
**Infant gender**		
Female		212 (51.7)
Male		198 (48.3)
**Weeks of gestation**	39.11 (1.90)	
**Infant age**		
≥3 months to <6 months		147 (35.9)
≥6 months to <9 months		133 (32.4)
≥9 months to <12 months		130 (31.7)
**Nocturnal sleep duration (min)**	611.04 (85.985)	
Missing data		1 (0.2)
**Night waking**	1.44 (1.59)	
**Method of falling asleep**		
While being fed		90 (22)
While being rocked		74 (18)
While being held		22 (5.4)
Alone in the crib		177 (43.2)
In the crib with parental presence		47 (11.5)
**IBQ-NEG**	3.36 (1.10)	

Note. City BiTS = City Birth Trauma Scale; EPDS = Edinburgh Postnatal Depression Scale; HADS-A = Anxiety subscale of the Hospital Anxiety and Depression Scale; IBQ-NEG = Negative Emotionality dimension of the Very Short Form of the Infant Behavior Questionnaire—Revised.

**Table 2 diagnostics-12-01625-t002:** Simple Linear Regression Models.

Model	Predictor	Dependent Variable	*n*	*β*	*R* ^2^	*F*	*p*
1	EPDS	Night waking	410	0.03	0.019	8.08	0.005
2	EPDS	Nocturnal sleep duration	409	−2.51	0.039	16.54	<0.001
3	HADS-A	Night waking	410	0.04	0.011	4.49	0.035
4	HADS-A	Nocturnal sleep duration	409	−2.59	0.016	6.77	0.010
5	City BiTS	Night waking	410	0.01	0.004	1.60	0.207
6	City BiTS	Nocturnal sleep duration	409	−0.80	0.010	4.17	0.042

Note. EPDS = Edinburgh Postnatal Depression Scale; HADS-A = anxiety subscale of the Hospital Anxiety and Depression Scale; City BiTS = City Birth Trauma Scale.

**Table 3 diagnostics-12-01625-t003:** Exploratory factor analysis with three predefined factors (*n* = 410).

Items	Depressive Profile	Birth Trauma Profile	Anxious Profile
**EPDS**			
1. Being able to laugh and see the funny side of things.	0.70		
2. Looking forward with enjoyment to things.	0.62		
3. Blaming oneself unnecessarily when things went wrong.	0.51		
4. Being anxious or worried for no good reason.			0.65
5. Feeling scared or panicky for no very good reason.			0.66
6. Things have been getting on top of oneself.	0.53		
7. Being so unhappy that one have had difficulty sleeping.	0.51		0.44
8. Feeling sad or miserable.	0.70		
9. Being so unhappy that one has been crying.	0.65		
10. The thought of harming oneself has occurred.	0.49		
**HADS-A**			
1. Feeling tense or angry.	0.64		
2. Feeling scared like something might happen to oneself.			0.64
3. Worrying.			0.66
4. Sitting restfully doing nothing and feeling calm.	0.41		
5. Feeling scared and having knots in one’s stomach.			0.64
6. Feeling restless and finding it difficult to stay in place.			
7. Often feeling panicky.			0.71
**City BiTS: birth-related symptoms**			
1. Recurrent unwanted memories of the birth.		0.71	
2. Bad dreams or nightmares about the birth.		0.49	
3. Flashbacks to the birth and/or reliving the experience.		0.49	
4. Getting upset when reminded of the birth.		0.79	
5. Feeling tense or anxious when reminded of the birth.		0.80	
6. Trying to avoid thinking about the birth.		0.77	
7. Trying to avoid things that remind me of the birth.		0.70	
8. Not able to remember details of the birth.			
9. Blaming myself or others for what happened during the birth.		0.62	
10. Feeling strong negative emotions about the birth.		0.70	
**City BiTS: General symptoms**			
11. Feeling negative about myself or thinking something awful will happen.	0.45		0.48
12. Lost interest in activities that were important to me.	0.65		
13. Feeling detached from other people.	0.69		
14. Not able to feel positive emotions.	0.70		
15. Feeling irritable or aggressive.	0.74		
16. Feeling self-destructive or acting recklessly.	0.50		
17. Feeling tense and on edge.	0.73		
18. Feeling jumpy or easily startled.			
19. Problems concentrating.	0.60		
20. Not sleeping well … not due to the baby’s sleep pattern.			0.45
Cronbach α, 95% CI	0.94, [0.93, 0.95]	0.87, [0.87, 0.90]	0.91, [0.89, 0.92]

Note. Only loading values > 0.40 are reported. City BiTS = City Birth Trauma Scale-French Version; EPDS = Edinburgh Postnatal Depression Scale; HADS-A = anxiety subscale of the Hospital Anxiety and Depression Scale.

**Table 4 diagnostics-12-01625-t004:** Significant moderated mediation models investigating relationships between maternal mental health symptom profiles and infant sleep.

Independent Variable	Dependent Variable	Mediator	Covariate	Moderator	ACME, 95% CI	*p*	Power Monte Carlo (Sobel)
**Model 1**							
Depressive profile	Night waking	IBQ-NEG	Method of falling asleep	Infant age: ≥3 months to <6 months	0.013, [0.006, 0.024]	<0.001	0.97 (0.82)
Depressive profile	Night waking	IBQ-NEG	Method of falling asleep	Infant age: ≥6 months to <9 months	0.006, [0.001, 0.014]	0.02	0.61 (0.47)
Depressive profile	Night waking	IBQ-NEG	Method of falling asleep	Infant age: ≥9 months to <12 months	0.009, [0.001, 0.024]	0.028	0.60 (0.48)
**Model 2**							
Depressive profile	Night waking	IBQ-NEG	Method of falling asleep	Low educational level	0.007, [0.002, 0.015]	0.004	0.73 (0.58)
Depressive profile	Night waking	IBQ-NEG	Method of falling asleep	High educational level	0.009, [0.004, 0.017]	<0.001	0.98 (0.87)
**Model 3**							
Depressive profile	Nocturnal sleep duration	IBQ-NEG	Method of falling asleep	Infant age: ≥6 months to <9 months	−0.296, [−0.677, −0.067]	0.012	0.57 (0.44)
**Model 4**							
Anxious profile	Night waking	IBQ-NEG	Method of falling asleep	Infant age: ≥3 months to <6 months	0.013, [0.004, 0.026]	<0.001	0.70 (0.55)
**Model 5**							
Anxious profile	Night waking	IBQ-NEG	Method of falling asleep	High educational level	0.009, [0.001, 0.021]	0.032	0.50 (0.48)

Note. ACME = average causal mediation effect; IBQ-NEG = negative emotionality subscale of the Very Short Form of the Infant Behavior Questionnaire-Revised. Non-significant moderated mediation models are not reported here. Post hoc power analyses for testing mediation effect based on 1000 Monte Carlo simulations and based on Sobel’s test [43,44].

## Data Availability

Data are available free of charge and without restriction from the open access repository Zenodo (https://doi.org/10.5281/zenodo.5070945 accessed on 4 May 2022) [39].

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
