# Peer review of "Maternal Mental Health Symptom Profiles and Infant Sleep: A Cross-Sectional Survey"

_diagnostics, 2022, doi:10.3390/diagnostics12071625_

Round 1

Reviewer 1 Report

Thank you for this very interesting paper. The topic is one less studied, so all your input is highly appreciated.

Good description of the theoretical background.

The methodology is clear.

Results are accurately analyzed.

The discussion part is comprehensive and comparison with other research results is offered.

The presentation of the limitations complete a well written paper.

Author Response

Dear Reviewer, we thank you very much for taking the time to review this manuscript, which we also believe is of importance, and we deeply appreciate your very supportive and positive comment. 

Reviewer 2 Report

 The distinct influence of different maternal mental health symptom profiles on infant sleep during the first year postpartum: a cross-sectional survey

1.      Overview:

a.       The paper is a thorough examination of the relationship between reported infant sleep and self-report maternal mental health. The paper is generally sound and well written.

2.      General/Larger Issues:

a.       I am concerned that while focusing on the mother, the infant sleep has become problematized, with night waking and short nocturnal sleep seen as “sleep problems” whereas they are actually often a part of the normal developmental variation – this could perhaps be addressed.

b.      I am concerned that the measure for infant sleep is reported by the mothers and it is known that this is inaccurate and biased. Mothers with mental health issues and children with problematic sleep will be systematically estimated differently when compared to more objective sleep estimates than other mothers and children which seems to potentially confound the results. Please explain how you can avoid this entanglement with this analysis ( see Quante M, Hong B, von Ash T, Yu X, Kaplan ER, Rueschman M, Jackson CL, Haneuse S, Davison K, Taveras EM, Redline S. Associations between parent-reported and objectively measured sleep duration and timing in infants at age 6 months. Sleep. 2021 Apr 9;44(4):zsaa217. doi: 10.1093/sleep/zsaa217. PMID: 33098646; PMCID: PMC8033447.)

3.      Title:

a.       I would suggest shortening the title: Maternal mental health symptom profiles and infant sleep (i.e distinct is implied, different can be assumed by the plural, study design doesn’t need to be in the title, infant basically means first 12 months). This is a stylistic preference so feel free to ignore.

4.      Abstract:

a.       Remove MMH acronym if you can as its not used elsewhere in the paper

b.      Missing : prior to numbered list

5.      Introduction:

a.       it would be good to include explicit hypotheses rather than just aims or, if not, justify why.

6.      Methods:

a.       Justify sample size in this section

b.      Be explicit re “good psychometric properties” – line 135

7.      Results:

a.       Table 3 – odd use of  “one’s” – a singular third person possessive determiner so can’t be used as in “7. Being so unhappy that one’s have had difficulty sleeping.” – rather – “7. Being so unhappy that one has had difficulty sleeping.”

8.      Discussion:

a.       none

9.  Conclusion - there isnt one but i think there should be as you finish on strengths and limitations which i feel is a weak ending- i suggest summarising the useful findings briefly - there are lots of implications for further research but how could this potentially inform things now
